# USP53 Exerts Tumor-Promoting Effects in Triple-Negative Breast Cancer by Deubiquitinating CRKL

**DOI:** 10.3390/cancers15205033

**Published:** 2023-10-18

**Authors:** Yi Liu, Wei Tang, Feng Yao

**Affiliations:** 1Department of Breast and Thyroid Surgery, Renmin Hospital of Wuhan University, Wuhan 430060, China; 2016302180150@whu.edu.cn; 2Department of Pediartrics, Renmin Hospital of Wuhan University, Wuhan 430060, China; xijun19@163.com

**Keywords:** triple-negative breast cancer, USP53, CRKL, deubiquitinase, epithelial–mesenchymal transition

## Abstract

**Simple Summary:**

Triple-negative breast cancer (TNBC) has a high recurrence rate and metastasis rate, and the prognosis for patients is poor. Identifying novel treatment avenues for TNBC is crucial for the treatment and prognosis of TNBC patients. Deubiquitinase deficiency is associated with the occurrence and progression of several diseases, including cancer. As a member of deubiquitinase, ubiquitin-specific peptidase 53 (USP53) is essential for the development of tumors such as hepatocellular carcinoma and renal clear cell carcinoma. However, the precise function of USP53 in TNBC remains unknown. Our study has identified USP53 as a molecular target that mediates the progression of TNBC. This not only deepens our understanding of the tumorigenesis and development of TNBC, but also provides new potential therapeutic targets for TNBC patients.

**Abstract:**

Breast cancer (BC) ranks in the top five malignant tumors in terms of morbidity and mortality rates. Among BC subtypes, TNBC has a high recurrence rate and metastasis rate and the worst prognosis. However, the exact mechanism by which TNBC develops is unclear. Here, we show that the deubiquitinase USP53 contributes to tumor growth and metastasis in TNBC. USP53 is overexpressed in TNBC, and this phenotype is linked to a poor prognosis. Functionally, USP53 promotes TNBC cell proliferation, migration, invasion, and epithelial–mesenchymal transition (EMT). More importantly, USP53 decreases the chemosensitivity of BC cells by enhancing v-crk sarcoma virus CT10 oncogene homologue (avian)-like (CRKL) expression. Mechanistically, USP53 directly binds CRKL to stabilize and deubiquitinate it, thereby preventing CRKL degradation. Overall, we discovered that USP53 deubiquitinates CRKL, encourages tumor development and metastasis, and reduces chemosensitivity in TNBC. These findings imply that USP53 might represent a new therapeutic target for the treatment of TNBC.

## 1. Introduction

Breast cancer (BC) has the highest incidence and mortality among women worldwide, according to the 2020 Global Cancer Statistics report. In 2020, BC was responsible for 2.3 million new cases and 685,000 fatalities [1]. A total of 15–20% of BC cases are TNBC, a highly proliferative and aggressive malignancy [2]. As the only BC subtype without targeted therapy, it has a high rate of recurrence and metastasis. Therefore, identifying novel treatment avenues for TNBC is crucial for the treatment and prognosis of TNBC patients.

Deubiquitinases have been the subject of a growing array of investigations in recent years. The occurrence and development of numerous diseases, including cancer, are linked to deubiquitinase dysfunction. USP53 has been found to promote osteogenic differentiation [3]. Mutations in the USP53 gene in humans can cause cholestasis [4,5,6], hearing loss [7], and other conditions. Additionally, USP53 is essential for the development of tumors. USP53 induces apoptosis of hepatocellular carcinoma cells through deubiquitination of cytochrome C [8] and is a tumor suppressive factor in esophageal carcinoma [9], lung adenocarcinoma [10], and renal clear cell carcinoma [11], but it has a significant inhibitory effect on the radiosensitivity of human cervical cancer [12]. The precise function of USP53 in TNBC, however, remains unknown.

CRKL belongs to the CRK adapter family and consists of modular SH2 and SH3 domains. CRKL is involved in epithelial morphogenesis [13], cell trait transformation [14], neural synapse formation [15,16], immune function [17], cell proliferation [18], cell adhesion [19,20], and EMT [21,22,23]. CRKL has been found to have a prominent ability to promote cell proliferation and metastasis in malignant tumors such as breast cancer [24,25,26] and cervical cancer [23]. CRKL has multiple forms of posttranslational modification in vivo. CRKL can be phosphorylated by BCR-Abl [27], and IFNα can promote the phosphorylation of CRKL [28]. C-cbl and Cbl-B are E3 ubiquitinating enzymes that both ubiquitinate CRKL and negatively regulate the CRKL–C3G complex [29,30]. However, little is known about the deubiquitinating enzyme of CRKL.

Epithelial cells undergo a process called EMT when they lose their polarity and start to resemble mesenchymal cells [31]. As a result, epithelial cells acquire significant levels of cellular plasticity, motility, invasiveness, and anti-apoptotic properties [32]. EMT has been demonstrated to be crucial for tumor formation in a number of malignancies, including lung cancer [33], hepatocellular carcinoma [34], and breast cancer [35]. The primary mechanisms by which EMT processes affect carcinoma cell cancer include activation of cancer stem cell properties, encouragement of cancer cell motility, invasion, and spread, and alteration of cancer treatment sensitivity [32,36]. To date, the regulatory function of USP53 in EMT has not been made explicit.

In this study, we verified that USP53 is a deubiquitinating enzyme for CRKL that directly binds, stabilizes, and deubiquitinates CRKL. Additionally, the knockdown of USP53 suppressed proliferation, EMT, and metastasis and enhanced the paclitaxel response of TNBC cells. In conclusion, USP53 plays an important role in TNBC progression by deubiquitinating CRKL.

## 2. Materials and Methods

### 2.1. Cell Culture and Transfection

The MDA-MB-231 and MDA-MB-468 cell lines were purchased from the American Type Culture Collection (Manassas, VA, USA) and were authenticated by STR profiling. Ninety percent DMEM (Gibco, Waltham, MA, USA) was used to culture all of the cells, with 10% fetal bovine serum (AusgeneX, Australia) and 1% penicillin/streptomycin (Biosharp, China) added as supplements. Cells were cultured in a 37 °C thermal cell incubator with 5% CO_2_.

### 2.2. Plasmids and siRNA

Flag-tagged USP53 was purchased from OBiO Technology (Shanghai, China), and siUSP53 was provided from GenePharma (Suzhou, China). The siRNA sequence was as follows: 5${}^{\prime }$-GTCACAAGGATGAATATAA-3${}^{\prime }$. All transfections performed using Lipo2000 (Invitrogen, Waltham, MA, USA) were performed according to the reagent instructions.

### 2.3. Reagents and Primary Antibodies

Cycloheximide (HY-12320), MG132 (HY-13259), and paclitaxel (HY-B0015) were provided by MCE. Anti-USP53 (A14353), anti-N-cadherin (A0433), anti-CRKL (A0511), anti-p-CRKL (AP0824), and anti-ubiquitin (A0162) were purchased from Abclonal. Anti-E-cadherin (20874-1-AP) was purchased from Proteintech. Anti-Flag (ab205606) was obtained from Abcam.

### 2.4. Western Blotting

SDS/PAGE gels were used for protein electrophoresis before being transferred to membranes. The membranes were blocked with 5% skim milk for two hours at room temperature and then incubated with the primary antibody overnight at 4 °C and the secondary antibody for one hour at room temperature. The images were taken on a Bio-Rad chemiluminescence system using ECL development reagents.

### 2.5. Immunoprecipitation

Primary antibodies were prepared with protein A/G agarose magnetic beads (sc-2003, Santa Cruz) and incubated at 4 °C for 4 h before centrifugation, and the supernatant was aspirated off for use. Cells were lysed with NP40. Subsequently, cell lysates were added to the primary antibody–agarose magnetic bead conjugations and incubated overnight at 4 °C. Loading buffer was added and boiled, and the supernatant samples were loaded onto SDS–PAGE gels. Binding proteins were detected using immunoprotein electrophoresis analysis.

### 2.6. Immunofluorescence Analysis

Cells were pierced with 0.2% Triton X-100 for 15 min at 4 °C after being fixed for 30 min with 4% paraformaldehyde. The cells were then treated with primary antibodies overnight at 4 °C, secondary antibodies for 30 min at room temperature in dark conditions, and DAPI dye for 3 min. The slides were microscopically observed.

### 2.7. Immunohistochemistry

The slides were dewaxed, submerged in citric acid antigen repair solution, heated in a microwave, and allowed to cool naturally. Slices were dipped in a 3% hydrogen peroxide solution and were incubated at room temperature for 25 min without exposure to light, overnight at 4 °C with the primary antibody, and for 50 min at room temperature with the secondary antibody before DAB chromogenic solution was added. The slices were stained with hematoxylin. According to the intensity (1: low; 2: weak; 3: moderate; 4: high) and the proportion of positive cells (1: 0–25%; 2: 26–50%; 3: 51–75%; 4: 76–100%), each slice was evaluated. Total score: intensity multiplied by percentage. A total score <8 was considered a low expression; otherwise, it was considered a high expression.

### 2.8. Quantitative Real-Time PCR

SYBR Green Master Mix (RK21203, Abclonal) was used to perform real-time fluorescence quantitative PCR (qRT-PCR) using an ABI 7500 real-time PCR machine. The primers were USP53 F: 5${}^{\prime }$-ATGGGTGTCAGATGCCAA-3${}^{\prime }$, R: 5${}^{\prime }$-CTGTGCTTCGGAAGATGAGA-3${}^{\prime }$; E-cadherin F: 5${}^{\prime }$-CGAGAGCTACACGTTCACGG-3${}^{\prime }$, R: 5${}^{\prime }$-GGGTGTCGAGGGAAAAATAGG-3${}^{\prime }$; CRKL F: 5${}^{\prime }$-GTGCTTATGACAAGACTGCCT-3${}^{\prime }$, R: 5${}^{\prime }$-CACTCGTTTTCATCTGGGTTT-3${}^{\prime }$; GAPDH F: 5${}^{\prime }$-TCACCACCATGGAGAAGGC-3${}^{\prime }$, R: 5${}^{\prime }$-GCTAAGCAGTTGGTGGTGCA-3${}^{\prime }$.

### 2.9. Cell Proliferation Assay

The cells were plated into 96-well plates for 24 h, and then the absorbance was measured at 450 nm after the CCK-8 reagent had been introduced for two hours. Measurements were made once a day for four consecutive days.

### 2.10. Migration and Invasion Assay

In the migration assay, the cells were fixed with 4% paraformaldehyde and stained with crystal violet before observation. For the invasion assay, 100 µL of 250 g/mL Matrigel was added to an 8 µm well filter. We added DMEM to hydrate the matrix glue before seeding the cells. The remaining steps followed the same procedures as the migration assay.

### 2.11. CHX Chase and Ubiquitination Assay

For the CHX assay, cells were transiently transfected for 48 h, then CHX was added, and after 0, 3, 6, and 9 h of treatment, the cells were harvested and lysed. Intracellular protein was measured using Western blotting. For the ubiquitination assay, cells were transiently transfected for 48 h with or without 50 µg/mL MG132 6 h before cell collection. CRKL antibodies were prepared and incubated with protein A/G agarose magnetic beads for 4 h at 4 °C. Cells were lysed using NP40, and cell lysates were added to anti-CRKL-agarose coupled magnetic beads and incubated for 6 h at 4 °C. Ubiquitination was detected using Western blotting.

### 2.12. Statistical Analysis

Student’s *t*-test or one-way ANOVA was used to determine the statistical significance between two or multiple groups, and the Spearman correlation test was used to perform the correlation, with nonsignificant (ns), *p* < 0.05 (*), *p* < 0.01(**), *p* < 0.001 (***), and *p* < 0.0001 (****). GraphPad Prism 8 was used to generate the figures and statistical analyses.

## 3. Results

### 3.1. USP53 Is Overexpressed in TNBC and Is Related to Poor Prognosis

To explore the expression of USP53 in TNBC, we measured USP53 protein expression and mRNA levels individually in five paired tumor and adjacent normal tissues of TNBC patients. We observed that the transcription and translation of USP53 were highly expressed in TNBC (Figure 1a,b). In addition, we selected breast tissues of TNBC patients for immunohistochemistry (IHC). In TNBC samples, USP53 protein expression was considerably higher than that in paired adjacent normal tissues (Figure 1c,d). Utilizing the PrognoScan database, survival analyses were carried out to determine if USP53 expression correlates with prognosis. We discovered that USP53 expression was linked to poor survival in breast cancer patients (Figure 1e). We also performed Kaplan–Meier survival analyses to assess the association between USP53 expression and survival in patients with TNBC. The findings demonstrated that patients with high USP53 expression had a considerably poorer survival rate than those with low USP53 expression (Figure 1f–h). Therefore, these findings imply that USP53 is crucial for the development and progression of TNBC. Next, we examined the protein expression level of USP53 in breast normal (MCF10A cell line) and cancer cell lines (T47D, MCF-7, MDA-MB-468, and MDA-MB-231 cell lines). We found that the protein level of USP53 was higher in breast cancer cell lines than in the normal cell line (Figure 1i). As the MDA-MB-231 cell lines and MDA-MB-468 cell lines belong to TNBC cell lines, we focus on their USP53 expression. We observed that the USP53 protein expression was high in MDA-MB-231 cells but low in MDA-MB-468 cells (Figure 1i). Therefore, in the following experiments, we chose MDA-MB-231 cells for USP53 knockdown experiments and MDA-MB-468 cells for USP53 overexpression experiments.

### 3.2. USP53 Promotes Cell Proliferation in TNBC

We used siUSP53 to knock down USP53 in MDA-MB-231 cells and the USP53 plasmid to overexpress USP53 in MDA-MB-468 cells to examine how USP53 affects the proliferation of TNBC cells. We used Western blotting and qRT-PCR analysis to verify the transfection effect. The USP53 protein expression and mRNA levels were reduced in MDA-MB-231 cells treated with siUSP53 (Figure 2a,b). Cell growth was detected by CCK-8 assay and colony formation assay. The results showed that knockdown of USP53 significantly reduced TNBC cell proliferation (Figure 2c,d,g). Conversely, The USP53 protein expression and mRNA levels were increased in MDA-MB-468 cells transfected with the USP53 plasmid (Figure 2e,f). Overexpression of USP53 accelerated TNBC cell proliferation (Figure 2f,h). To assess whether USP53 promotes the growth of breast tumors in vivo, we constructed an MDA-MB-231 cell line with stable shUSP53 expression and verified the USP53 expression levels through Western blotting (Figure 2i). We injected 5 × $10^{6}$ cells into the axilla of five female BALB/C nude mice in the indicated groups individually and fed the mice for one month. We found that the growth rate of TNBC cells in the USP53 knockdown group was much lower than that in the control group (Figure 2j–l). Ki-67 IHC staining was conducted on mouse mammary tissues to further evaluate the proliferation ability of mouse mammary tumors. As shown, the knockdown of USP53 significantly reduced the proliferation of mouse mammary tumors (Figure 2m). IHC staining of tissues from TNBC patients revealed a significant correlation between USP53 and Ki-67 expression levels (Figure 2o,p). Our results collectively indicate that USP53 is critical for TNBC carcinogenesis and proliferation.

### 3.3. USP53 Promotes TNBC Cell Metastasis

Next, we examined the role of USP53 in the metastasis of TNBC. We performed transwell assays after transfection with siControl or siUSP53 in MDA-MB-231 cells and the vector control or USP53 plasmid in MDA-MB-468 cells to evaluate the possible impact of USP53 on tumor migration and invasion in vitro. Knockdown of USP53 reduced the proportions of migrated and invaded MDA-MB-231 cells (Figure 3a,b). Similarly, overexpression of USP53 raised the proportions of migrated and invaded MDA-MB-468 cells (Figure 3c,d). Next, we examined the protein expression levels of typical markers of EMT by Western blot analysis. Knockdown of USP53 decreased Snail1 and N-cadherin expression levels and increased E-cadherin expression levels in MDA-MB-231 cells (Figure 3e). The results were reversed upon USP53 overexpression in MDA-MB-468 cells (Figure 3eg). We further measured the mRNA levels of E-cadherin, and the variations were consistent with the Western blotting (Figure 3f,h). To ulteriorly verify these results, we conducted N-cadherin and DAPI immunofluorescence staining in siControl- and siUSP53-treated MDA-MB-231 cells and in the vector control or USP53 plasmid MDA-MB-468 cells. Consistently, compared to the control group, the expression of N-cadherin was dramatically decreased upon USP53 knockdown in MDA-MB-231 cells and increased upon USP53 overexpression in MDA-MB-468 cells (Figure 3i,j). IHC staining for E-cadherin was performed in xenograft tissues of subcutaneously inoculated nude mice, and the mice in the USP53 knockdown group had higher protein expression levels of E-cadherin (Figure 3k). Additionally, IHC staining of TNBC patient tissues confirmed a negative correlation between the expression of USP53 and E-cadherin (Figure 3l,m). Taken together, USP53 promotes metastasis and EMT in TNBC.

### 3.4. USP53 Regulates the Protein Level of CRKL in TNBC

We explored the STRING database to determine the precise mechanism by which USP53 controls EMT and discovered that USP53 can directly bind CRKL (Figure 4a). CRKL is known to promote metastasis and the EMT process in cancers such as hepatocellular carcinoma, cervical cancer, and gastric cancer and correlates with a poor prognosis [21,22,23]. To assess whether USP53 is involved in the metastatic progression of TNBC by regulating CRKL, we first examined p-CRKL and CRKL protein levels by Western blotting. We found that p-CRKL and CRKL protein levels were visibly reduced in siUSP53-treated MDA-MB-231 cells and increased in USP53 plasmid-treated MDA-MB-468 cells (Figure 4b,d). The ratio of p-CRKL/CRKL was also reduced in siUSP53-treated MDA-MB-231 cells and upregulated in USP53 plasmid-treated MDA-MB-468 cells (Figure 4c,e). To further examine CRKL expression, we performed a CRKL immunofluorescence assay. USP53 knockdown reduced CRKL expression in MDA-MB-231 cells, and USP53 overexpression elevated the expression of CRKL in MDA-MB-468 cells (Figure 4f,g). Next, we performed IHC staining for CRKL in xenograft tissues of subcutaneously inoculated nude mice. CRKL protein levels were evidently decreased in the USP53 knockdown group compared with the control group (Figure 4h). Furthermore, IHC staining of TNBC patient tissues revealed that USP53 and CRKL expression was positively correlated (Figure 4i,j). Taken together, USP53 promotes the process of EMT in TNBC partly by regulating CRKL protein expression.

### 3.5. USP53 Binds, Stabilizes, and Deubiquitinates CRKL

Next, we sought to determine how USP53 regulates CRKL protein expression, whether through transcriptional regulation or posttranslational modification. The qRT-PCR results revealed that CRKL mRNA levels in MDA-MB-231 cells with USP53 knockdown and MDA-MB-468 cells with USP53 overexpression did not differ from those in the control group. (Figure 5a,b). Therefore, we hypothesized that USP53 is involved in the regulation of CRKL protein expression through posttranslational modification. The proteasome inhibitor MG132 significantly attenuated the downregulation of CRKL induced by USP53 knockdown, whereas CRKL expression generated by USP53 knockdown was not significantly impacted by the autophagy inhibitor 3-MA (Figure 5c,d). This indicates that USP53 controls CRKL through the proteasome mechanism as opposed to the autophagy pathway. To further confirm these findings, cycloheximide (CHX) chase analysis was performed. Knockdown of USP53 significantly shortened the half-life of CRKL degradation (Figure 5e,f). As a deubiquitinase, USP53 may control CRKL by deubiquitinating CRKL. Using a ubiquitin assay, we observed markedly increased ubiquitination levels of CRKL in cells with USP53 knockdown and decreased ubiquitination levels of CRKL in cells with USP53 overexpression (Figure 5g,h). Next, we wanted to determine whether USP53 deubiquitinates K48-linked or K63-linked ubiquitin chains of CRKL. The results showed that K48 ubiquitinated CRKL was significantly increased after USP53 knockdown, suggesting that USP53 could deubiquitinate K48 ubiquitin-linked CRKL but not K63 ubiquitin-linked CRKL (Figure 5i). We further overexpressed HA-ubiquitin, HA-ubiquitin (K48R), or HA-ubiquitin (K63R) while knocking down USP53, and the results further validated that USP53 acts on the K48 ubiquitin linkage of CRKL but not the K63 ubiquitin linkage (Figure 5j). To evaluate whether USP53 regulates the deubiquitination of CRKL through direct binding or indirect pathways, immunofluorescence colocalization and immunoprecipitation analyses were performed. The immunofluorescence colocalization results showed that USP53 and CRKL were mainly located in the cytoplasm of MDA-MB-231 and MDA-MB-468 cells (Figure 5k). Coimmunoprecipitation analysis showed that endogenous USP53 could bind CRKL (Figure 5l,m). Taken together, USP53 can directly bind, stabilize, and deubiquitinate CRKL.

### 3.6. Knockdown of CRKL Can Reverse Cell Proliferation, Migration, and Invasion Induced by USP53 Upregulation

Because upregulation of USP53 promoted proliferation, migration, invasion, and EMT in TNBC, we investigated whether this was due to the regulation of CRKL by USP53. Vector control, the USP53 plasmid, or the USP53 plasmid with siCRKL were transfected into MDA-MB-468 cells. MDA-MB-468 cell proliferation, migration, and invasion were all facilitated by USP53 overexpression (Figure 6a–f). However, they were partially rescued by the knockdown of CRKL (Figure 6a–f). To explore whether the effect of USP53 on EMT is due to the regulation of CRKL, we further measured related protein levels. Knockdown of CRKL restored the protein levels of Snail1, N-cadherin, E-cadherin, p-CRKL, and CRKL (Figure 6g–i). Consistently, the knockdown of CRKL restored the expression levels of N-cadherin and CRKL, as shown by immunofluorescence staining (Figure 6j,k). Therefore, the effect of USP53 on TNBC cell proliferation and metastasis is at least partially mediated by the regulation of CRKL. USP53 may regulate the proliferation and metastasis of TNBC by deubiquitinating CRKL.

### 3.7. USP53 Reduces Chemosensitivity in TNBC

There are two studies that suggest a link between EMT and chemotherapy sensitivity. EMT can promote lung metastasis of breast cancer that reoccurs after chemotherapy. Overexpression of miR-200 attenuated this chemoresistance [37]. In addition, EMT has also been shown to induce chemotherapy resistance in pancreatic cancer [38]. In our recent study, we showed that USP53 can regulate EMT in TNBC. In addition, it has been found that USP53 can regulate the radiosensitivity of cervical cancer [12]. Therefore, we wondered whether USP53 could affect chemoresistance in TNBC. In the treatment of BC, paclitaxel (PTX) is a popular chemotherapeutic medication. We used different doses of PTX to determine the optimal concentration of PTX in TNBC cells. According to the results (Figure 7a,b), USP53 expression showed a dose-dependent relationship with PTX. Next, we treated MDA-MB-231 cells with PTX (50 µM) following transfection of MDA-MB-231 cells with siControl or siUSP53 and MDA-MB-468 cells with the vector control or the USP53 plasmid. Cell proliferation assays showed that USP53 knockdown inhibited cell proliferation and increased chemosensitivity to PTX in MDA-MB-231 cells (Figure 7c). Consistently, USP53 overexpression promoted cell proliferation and decreased chemosensitivity to PTX in MDA-MB-468 cells (Figure 7d). Further flow cytometry analysis proved that siUSP53 MDA-MB-231 cells generated by PTX had a considerably higher cell apoptosis percentage than the control group (Figure 7e,f), whereas overexpression of USP53 in MDA-MB-468 cells decreased chemosensitivity to PTX (Figure 7g,h). Interestingly, this effect could be rescued by knocking down CRKL (Figure 7i,j). To further verify our results, Western blotting precipitation was used to detect USP53 and PARPα protein levels. PARPα expression in the knockdown group was significantly decreased under paclitaxel induction, whereas PARPα expression in the USP53 overexpression group was not significantly increased (Figure 7k,l). This suggests that knockdown of USP53 may increase the chemotherapy sensitivity of TNBC cells partly by promoting autophagy. In conclusion, overexpression of USP53 decreases chemosensitivity in TNBC.

## 4. Discussion

Deubiquitination is involved in regulating various cellular activities, including tumor growth, migration, invasion, and EMT progression. Deubiquitinating enzymes (DUBs) are crucial to the generation and progression of BC. For example, ATXN3L and USP3 mediate the deubiquitination of KLF5 and thus regulate the proliferation of BC cells [39,40]. USP27X promotes EMT progression in tumors by deubiquitinating and stabilizing Snail1 [41]. Studies have discovered that USP53 is crucial for malignancies. After activation by H3K27 acetylation, USP53 inhibits the progression of esophageal cancer through the AMPK pathway [9]. Lung adenocarcinoma cells are regulated by USP53, which deubiquitinates FKBP51, dephosphorylates AKT1, and controls glycolysis and apoptosis [10]. USP53 also deubiquitinates cytochrome c and promotes apoptosis of hepatoma cells [8]. Additionally, by combining with DDB2, USP53 improved the radiotherapy sensitivity of cervical squamous cell carcinoma cells [12]. However, whether USP53 plays a role in TNBC is not known. In this study, we found that USP53 promoted the growth and metastasis of TNBC. Furthermore, cells with low USP53 expression showed better sensitivity to chemotherapy. In short, USP53 acts as an oncogene in TNBC.

CRKL is an oncogenic adaptor protein containing SH2 and SH3 domains [42,43] that controls cell proliferation [18], cell adhesion [19,20], cell metastasis, EMT [21,22,23], and tumor formation [18]. In chronic myeloid leukemia, CRKL is a crucial BCR/ABL substrate that can bind to both c-ABL and BCR/ABL. CRKL is upregulated in many malignancies, such as gastric cancer [23], hepatocellular carcinoma [10,44,45], lung cancer [46,47], and cervical cancer [23], and correlates with a poor disease prognosis. CRKL is highly expressed in BC and promotes cell growth and invasion [48]. By controlling miR-200c targeting CRKL, histone deacetylase inhibitors can suppress BC cell proliferation and invasion [49]. C3G, SOS, PI3K, c-ABL, and BCR/ABL bind to the amino-terminal SH3 domain of CRKL. CBL, HEF1, CAS, or paxillin can be bound by the SH2 domain of CRKL [42,43,50]. CRKL regulates cell proliferation, cell adhesion, cell metastasis, EMT, and tumor formation. However, the precise degradation process of CRKL is not clear. Here, we found that CRKL can be deubiquitinated by USP53. Notably, USP53 is the first DUB found to be able to deubiquitinate CRKL, so our study has important physiological and pathological implications. Our study shows that USP53 can directly bind, stabilize, and deubiquitinate CRKL and further promote proliferation, migration, invasion, and EMT in TNBC.

Recent research has demonstrated that CRKL can control EMT, impacting the development and spread of malignancies. By interacting with AFAP1-AS1, CRKL encourages the growth and EMT of hepatocellular carcinoma cells [21]. In addition, miR-429 inhibits CRKL expression. Knockdown of miR-429 in gastric cancer cells increased CRKL protein levels and promoted EMT through the Akt signaling pathway [22]. SASH1 can bind to the oncoprotein CRKL and inhibit CRKL-mediated SRC kinase activation, thereby inhibiting EMT [51]. However, CRKL is overexpressed in bladder cancer and inhibits tumor cell proliferation and migration [52]. According to one study, HDAC inhibitors can delay BC progression by upregulating miR-200c and downregulating CRKL [49]. The function of CRKL in TNBC cell EMT is not yet well understood. In this study, we demonstrated that CRKL controls breast cancer cell EMT by elevating Snail1 and N-cadherin protein levels and decreasing E-cadherin expression.

Ubiquitination and deubiquitination are finely regulated in vivo to control cell growth and control the occurrence and development of tumors [53,54]. The overexpression of USP53 in TNBC promotes the deubiquitination of CRKL and tumor growth and metastasis in TNBC. In addition, USP53 reduces sensitivity to chemotherapy in TNBC. However, further study is needed to determine the sites by which USP53 interacts with CRKL and identify strategies that can be applied to regulate the degree of CRKL deubiquitination by USP53. In this study, we did not further evaluate the domains involved in CRKL and USP53 function, which need further exploration. In addition, whether other DUBs can deubiquitinate CRKL remains to be determined.

## 5. Conclusions

In summary, we found that USP53 can deubiquitinate CRKL, thus regulating tumor proliferation, metastasis, and sensitivity to chemotherapy in TNBC (Figure 7m). Notably, USP53 is the first DUB found to deubiquitinate CRKL. These results not only deepen our understanding of the tumorigenesis and development of TNBC but also provide new potential therapeutic targets for TNBC patients.

## Figures and Tables

**Figure 1 cancers-15-05033-f001:**
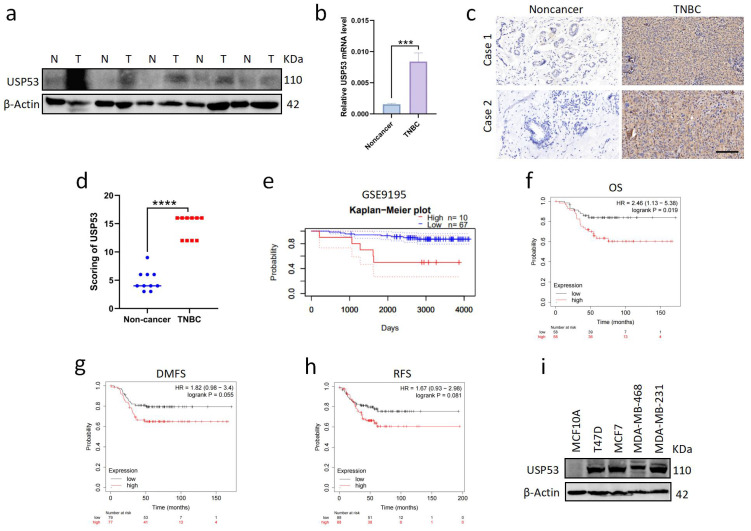
USP53 is overexpressed in TNBC and is related to a poor prognosis. (**a**) Protein expression of USP53 in five paired tumor and adjacent normal tissues of five TNBC patients. (**b**) The mRNA levels of USP53 in five paired tumor and adjacent normal tissues of five TNBC patients. (**c**,**d**) Typical IHC staining of USP53 in paired tumor and adjacent normal tissues. IHC staining was performed on 30 pairs of TNBC tissues, and a representative image of two pairs is shown. Scatter dot plot analysis of USP53 expression is shown (by *t*-test) on the right. Scale bar: 100 µm. (**e**) High USP53 expression correlates with poor overall survivals (OS) in breast cancer patients from the PrognoScan cohort. (**f**–**h**) High USP53 expression correlates with poor overall survival, distant metastasis free survival (DMFS), and recurrence free survival (RFS) in TNBC patients from the TCGA cohort. (**i**) Representative Western blot of USP53 expression in MCF10A, T47D, MCF-7, MDA-MB-468, and MDA-MB-231 cells; β-Actin served as a loading control. *** *p* < 0.001, **** *p* < 0.0001. Original western blots are presented in Appendix A.

**Figure 2 cancers-15-05033-f002:**
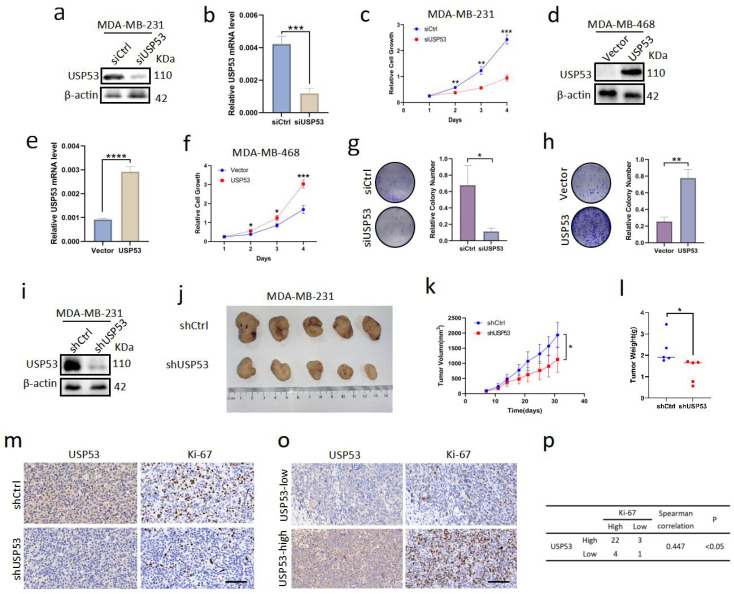
USP53 promotes cell proliferation in TNBC. (**a**) USP53 knockdown in MDA-MB-231 cells was confirmed by Western blotting. (**b**) The mRNA levels of USP53 in MDA-MB-231 cells transfected with siControl or siUSP53. (**c**) CCK-8 assay was performed following transfection. MDA-MB-231 cells were transfected with siControl or siUSP53. (**d**) USP53 overexpression in MDA-MB-468 cells was confirmed by Western blotting. Vector alone was used as the negative control for USP53. (**e**) The mRNA levels of USP53 in MDA-MB-468 cells transfected with the vector control or the USP53 plasmid. (**f**) CCK-8 assay was performed following transfection. MDA-MB-468 cells were transfected with vector or USP53. (**g**) Colony formation assay of siControl- and siUSP53-treated MDA-MB-231 cells. (**h**) Colony formation assay of vector- and USP53 plasmid-treated MDA-MB-468 cells. (**i**) The USP53 expression of MDA-MB-231 cells transfected with shControl or shUSP53 was verified by Western blotting. (**j**) The xenograft tumors of nude mice in the indicated groups after MDA-MB-231 cells injection for one month. (**k**,**l**) The volumes and weights of the mouse xenograft tumors in the indicated groups. (**m**) Typical IHC staining of USP53 in xenograft tissues of subcutaneously inoculated nude mice. Scale bar: 100 µm. (**o**) Representative IHC images of USP53 and Ki-67 in TNBC specimens. Scale bar: 100 µm. The correlation between USP53 and Ki-67 expression in TNBC specimens is shown in (**p**) (by the Spearman correlation test). * *p* < 0.05, ** *p* < 0.01, *** *p* < 0.001, **** *p* < 0.0001. Original western blots are presented in Appendix A.

**Figure 3 cancers-15-05033-f003:**
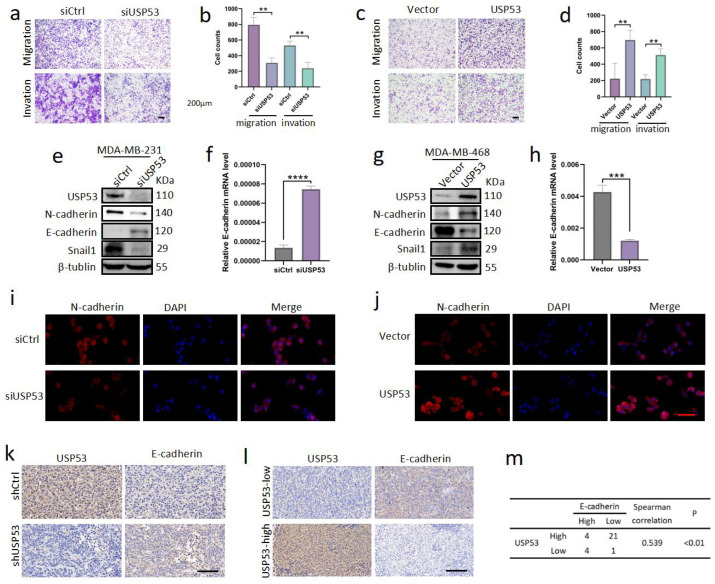
USP53 promotes the metastasis of TNBC cells. (**a**,**b**) Knockdown of USP53 reduced the proportions of migrated and invaded MDA-MB-231 cells. Scale bar: 200 µm. (**c**,**d**) Overexpression of USP53 increased the proportions of migrated and invaded MDA-MB-468 cells. Scale bar: 200 µm. (**e**,**g**) Protein levels of Snail1, N-cadherin, E-cadherin, and USP53 were measured by Western blot analysis following transfection with siControl or siUSP53 in MDA-MB-231 cells and the vector control or USP53 plasmid in MDA-MB-468 cells. (**f**,**h**) The mRNA levels of USP53 in MDA-MB-231 cells transfected with siControl or siUSP53 and in MDA-MB-468 cells transfected with the vector control or USP53 plasmid. (**i**,**j**) Representative pictures of N-cadherin and DAPI immunofluorescence staining in siControl or siUSP53-treated MDA-MB-231 cells and in the vector control or USP53 plasmid MDA-MB-468 cells. Scale bar: 50 µm. (**k**) Typical IHC staining of E-cadherin in xenograft tissues of subcutaneously injected nude mice. Scale bar: 100 µm. (**l**) Representative IHC images of USP53 and E-cadherin in TNBC specimens. Scale bar: 100 µm. The correlation between USP53 and E-cadherin expression in TNBC specimens is shown in (**m**) (by the Spearman correlation test). ** *p* < 0.01, *** *p* < 0.001, **** *p* < 0.0001. Original western blots are presented in Appendix A.

**Figure 4 cancers-15-05033-f004:**
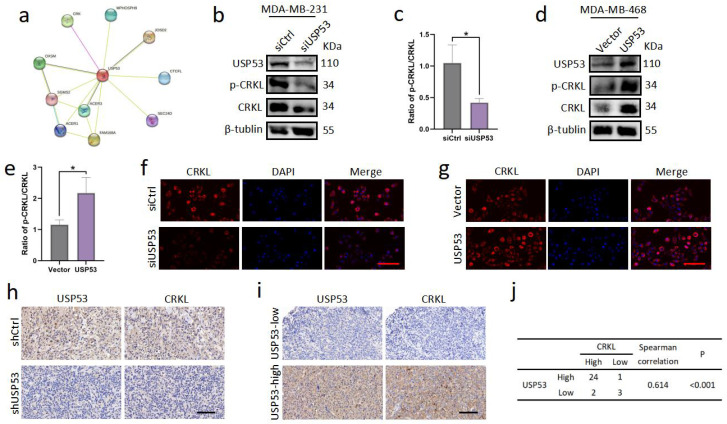
USP53 regulates the protein expression of CRKL in TNBC. (**a**) Network diagram illustrating the interaction of USP53 and CRKL. The network was created using the STRING database. The pink line represents the interaction that was experimentally determined. (**b**,**c**) Western blotting and statistics results of p-CRKL and CRKL following transfection with siControl or siUSP53 in MDA-MB-231 cells. (**d**,**e**) Western blotting and statistics results of p-CRKL and CRKL following transfection with the vector control or USP53 plasmid in MDA-MB-468 cells. (**f**,**g**) Representative pictures of CRKL and DAPI immunofluorescence staining in siControl- and siUSP53-treated MDA-MB-231 cells and in the vector control or USP53 plasmid MDA-MB-468 cells. Scale bar: 50 µm. (**h**) Typical IHC staining of CRKL in xenograft tissues of subcutaneously inoculated nude mice. Scale bar: 100 µm. (**i**) Representative IHC images of USP53 and CRKL in TNBC specimens. Scale bar: 100 µm. The correlation between USP53 and CRKL expression in TNBC specimens is shown in (**j**) (by the Spearman correlation test). * *p* < 0.05. Original western blots are presented in Appendix A.

**Figure 5 cancers-15-05033-f005:**
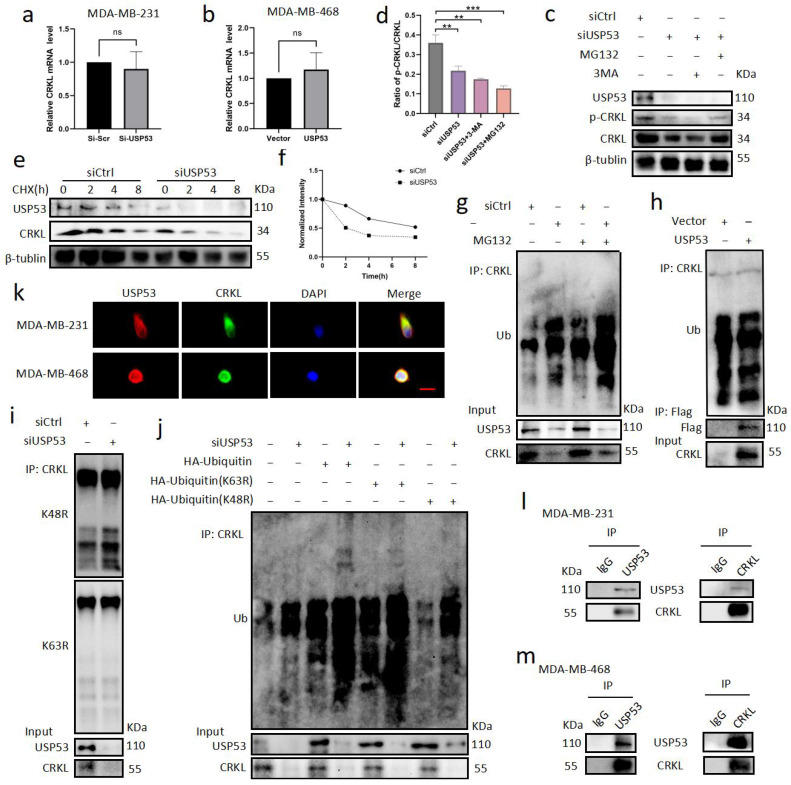
USP53 binds, stabilizes, and deubiquitinates CRKL. (**a**,**b**) Knockdown of CRKL in MDA-MB-231 cells and overexpression of CRKL in MDA-MB-468 cells did not change the relative CRKL mRNA levels according to qRT-PCR. (**c**,**d**) MDA-MB-231 cells were transfected with siControl or siUSP53 for 48 h and then treated with MG-132 (50 µM) or 3-MA (10 µM) for 6 h before harvesting. Western blotting and statistic results of p-CRKL and p-CRKL were shown. (**e**) The expression levels of USP53 and CRKL were analyzed by Western blot analysis in siControl and csiUSP53 MDA-MB-231 cells treated with cycloheximide (50 MM) for the indicated times. Quantification of the protein stability of CRKL is shown in (**f**). (**g**) MDA-MB-231 cells were transfected with siControl or siUSP53 for 48 h. Cells were treated with MG132 (10 µM) for 6 h before harvesting. Ubiquitinated CRKL, USP53, and CRKL protein levels were measured by Western blotting. (**h**) MDA-MB-468 cells were transfected with vector control or Flag-USP53 for 48 h. Ubiquitinated CRKL, Flag, and CRKL protein levels were measured by Western blotting. (**i**) MDA-MB-231 cells were transfected with siControl or siUSP53 for 48 h. K48-ubiquitinated CRKL, K63-ubiquitinated CRKL, USP53, and CRKL protein levels were measured by Western blotting. (**j**) MDA-MB-231 cells were transfected with siControl or siUSP53 and cotransfected with HA-ubiquitin, HA-ubiquitin (K63R), or HA-ubiquitin (K48R) plasmids for 48 h. Ubiquitinated CRKL, USP53, and CRKL protein levels were measured by Western blotting. (**k**) Immunofluorescence and DAPI staining of MDA-MB-231 and MDA-MB-468 cells coexpressing endogenous USP53 and CRKL. Scale bar: 50 µm. (**l**,**m**) Coimmunoprecipitation of endogenous USP53 and CRKL proteins in MDA-MB-231 (**l**) and MDA-MB-468 cells (**m**). ** *p* < 0.01, *** *p* < 0.001. Original western blots are presented in Appendix A.

**Figure 6 cancers-15-05033-f006:**
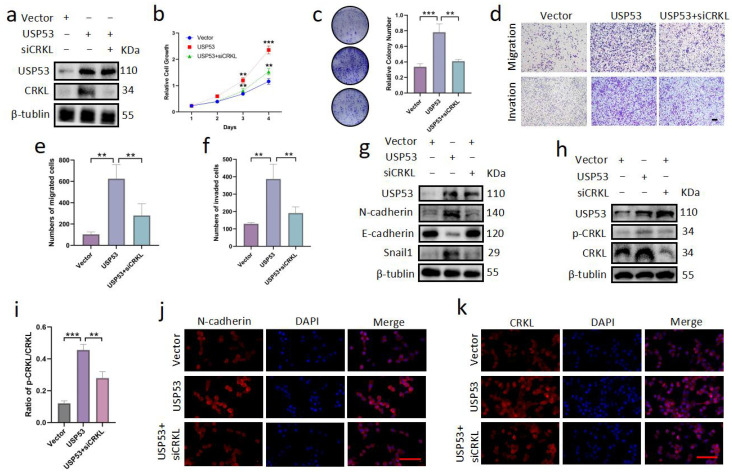
Knockdown of CRKL can reverse cell proliferation, migration, and invasion induced by USP53 downregulation. (**a**) USP53 overexpression and CRKL knockdown in MDA-MB-468 cells were confirmed by Western blotting. (**b**) CCK-8 assay was performed in MDA-MB-468 cells transfected with vector control, USP53 plasmid, or cotransfected with USP53 plasmid and siCRKL for 48 h. (**c**) Colony formation assay was performed in MDA-MB-468 cells transfected with vector control, USP53 plasmid, or cotransfected with USP53 plasmid and siCRKL for 48 h. (**d**–**f**) Migration and invasion assays were performed in MDA-MB-468 cells transfected with vector control, USP53 plasmid, or cotransfected with USP53 plasmid and siCRKL for 48 h. Scale bar: 200 µm. (**g**) The protein levels of Snail1, N-cadherin, and E-cadherin in MDA-MB-468 cells were measured by Western blotting. (**h**,**i**) Western blotting and statistics results of p-CRKL and CRKL in MDA-MB-468 cells in the indicated groups. (**j**,**k**) Representative pictures of CRKL and DAPI immunofluorescence staining in MDA-MB-468 cells transfected with vector control, USP53 plasmid, or cotransfected with USP53 plasmid and siCRKL for 48 h. Scale bar: 100 µm. ** *p* < 0.01, *** *p* < 0.001. Original western blots are presented in Appendix A.

**Figure 7 cancers-15-05033-f007:**
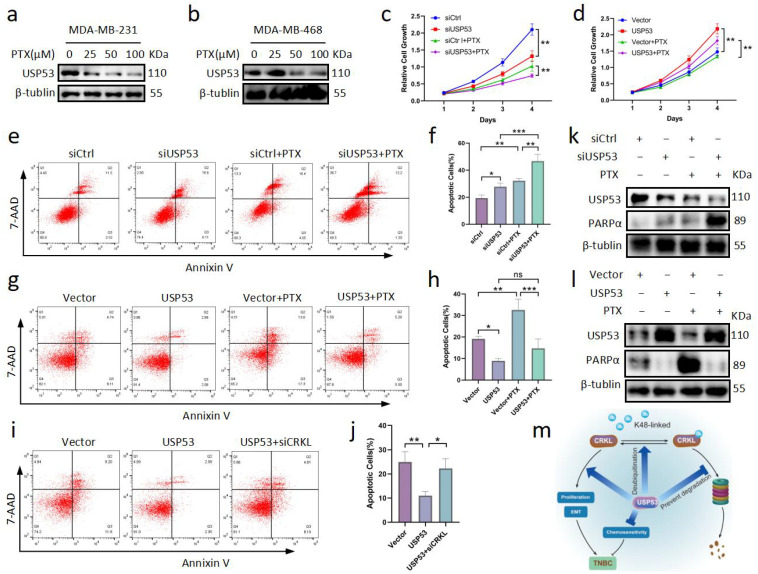
USP53 reduces chemosensitivity in TNBC. (**a**,**b**) USP53 expression in MDA-MB-231 and MDA-MB-468 cells by Western blot analysis at different concentrations of 0, 25, 50, 75, and 100 µM for 24 h. (**c**) CCK-8 assay was performed in MDA-MB-231 cells. Cells were transfected with siControl or siCRKL for 48 h and then treated with PTX (50 µM) for 48 h. (**d**) CCK-8 assay was performed in MDA-MB-468 cells. Cells were transfected with vector control or USP53 plasmid for 48 h and then treated with PTX (50 µM) for 48 h. (**e**,**f**) Flow cytometry apoptosis analysis was measured by Annexin V/7-AAD staining followed by flow cytometry in MDA-MB-231 cells. (**g**,**h**) Flow apoptosis analysis was measured by Annexin V/7-AAD staining followed by flow cytometry in MDA-MB-468 cells. (**i**,**j**) Flow apoptosis analysis was measured by Annexin V/7-AAD staining followed by flow cytometry in MDA-MB-468 cells transfected with vector control, USP53 plasmid, or cotransfected with USP53 plasmid and siCRKL for 48 h. (**k**,**l**) The protein levels of USP53 and PARPα were measured by Western blotting. (**m**) Schematic representation of the mechanism by which USP53 deubiquitinates CRKL in TNBC. * *p* < 0.05, ** *p* < 0.01, *** *p* < 0.001. Original western blots are presented in Appendix A.

## Data Availability

The corresponding author will provide the datasets created during and/or analyzed during the current investigation upon reasonable request.

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
