# Peer review of "USP53 Exerts Tumor-Promoting Effects in Triple-Negative Breast Cancer by Deubiquitinating CRKL"

_cancers, 2023, doi:10.3390/cancers15205033_

Round 1

Reviewer 1 Report

With pleasure, I read the paper titled “USP53 exerts tumor-promoting effects in triple-negative breast cancer by deubiquitinating CRKL”. The topic is clinically relevant and of importance to the readers of the Cancers journal. Overall, the manuscript reads good and has good flow of ideas, up-to-date citations, and good summary of data using tables and figures. A major strength of the article is being among the first-ever studies to examine the role of USP53 in TNBC. I have the following comments:

Figure 1. Have you examined the mRNA level of USP53 between normal and cancerous tissues? Also, have you examined the protein level of USP53 between normal and cancerous tissues using western blot? For panels d-f, please spell out the abbreviations. In panel g, it would be great to include non-cancerous cell lines as control. Out of curiosity, is the expression of USP53 differentially expressed between TNBC and other types of breast cancer?

Figure 2. The western blot does not show good knockdown of USP53 in MDA-MB-231 cells, and therefore, the data of proliferation and colony formation assays cannot be trusted. The authors may need to consider other siRNA oligos or even use shRNAs. Most importantly, have you examined the mRNA of USP53 after siRNA knockdown? The colony formation assay pictures are fine, but not the best. Similarly, the data for overexpression has a big question mark as there is n dramatic increase in expression after overexpression. I recommend authors validate the overexpression by at least examining also the mRNA level of USP53. Again, the authors need to establish that the in-vivo shRNA knockdown has actually proper knockdown by performing western blot, before executing the xenograft studies. Please indicate in legends the number of replicates and mice.

Figure 3. The knockdown efficiency here is good but did not work well in figure 2. Could you please explain the discrepancy.  Have you examined whether the EMT changes happen also at the transcriptional mRNA levels and whether that are matched with the protein levels?

Figure 4. it would be interesting to examine the mRNA level of CRKL as well upon siRNA knockdown and overexpression. It is important to quantify the bands for p-CRKL and total-CRKL as in panel c, the data suggest only equal upregulation of total and phosphor levels.

Figure 5. Sometimes your knockdown is good and sometimes it is not good. I don’t understand why!

Figure 6. For panel c, was the colony formation done in triplicates? For panels g and h, I am assuming the last lane should be positive for USP53 and siCRKL; please correct.

General questions that will substantially enhance the quality of your research. Have you examined the phenotypic effects of genetic inhibition of USP53 on DNA damage, cell cycle, and differentiation. Have you examined the RNA-seq profile of USP53 knockdown and exploring the enriched and deleted signaling pathways in an unbiased fashion? Is there a pharmacologic inhibitor of USP53 to mimic siRNA knockdown of USP53? Have you examined if pharmacological inhibition of USP53 show similar phenotype of increased siRNA knockdown of USP53? The study will become more significant if in-vivo xenograft data are included.

Minor English polishing

Reviewer 2 Report

The manuscript entitled “USP53 exerts tumor-promoting effects in triple-negative breast cancer by deubiquitinating CRKL” provides new insights about the tumorigenesis and development of TNBC. However, the manuscript needs major revision to make the suitable for publication.

1.      All Western blots in MS should be quantified.

2.      In would be worthy to confirm the effect of over expression of USP53 in in vivo.

3.      In figure, confirm the legend for (c,d). Overexpression of USP53 reduced the proportions of migrated and invaded MDA-MB-468 cells it should be Overexpression of USP53 reduced the proportions of migrated and invaded MDA-MB-468

4.      Raw Western blots are confusing, please label or write to full membrane and indicate bands used in main figure. It’s hard to trust the Western blot images as authors have cropped wherever they want in full blot.

5.      Authors should also use another triple negative cancer cell line to exclude cell line specific effects. 

Round 2

Reviewer 1 Report

For the most part, the authors did a great job by addressing the concerns and revising their manuscript, which now is scientifically stronger and intellectually robust. Nonetheless, some experiments would have added a better story. The limitation of funds to conduct some RNA-seq data is well-acknowledged. Overall, the manuscript can be accepted in its current format now.

Minor to none English editing is required.

Reviewer 2 Report

MS looks better after revision. I would recommend for publication.